# Microbial Spectrum and Antibiotic Resistance in Patients Suffering from Penetrating Crohn’s Disease

**DOI:** 10.3390/jcm11154343

**Published:** 2022-07-26

**Authors:** Simon Kusan, Güzin Surat, Matthias Kelm, Friedrich Anger, Mia Kim, Christoph-Thomas Germer, Nicolas Schlegel, Sven Flemming

**Affiliations:** 1Department of General, Visceral, Transplantation, Vascular and Pediatric Surgery, Center of Operative Medicine (ZOM), University Hospital of Wuerzburg, 97080 Wuerzburg, Germany; simon.kusan@stud-mail.uni-wuerzburg.de (S.K.); kelm_m@ukw.de (M.K.); anger_f@ukw.de (F.A.); mia.kim@muenchen-klinik.de (M.K.); germer_c@ukw.de (C.-T.G.); schlegel_n@ukw.de (N.S.); 2Unit for Infection Control and Antimicrobial Stewardship, University Hospital of Wuerzburg, 97080 Wuerzburg, Germany; surat_g@ukw.de

**Keywords:** Crohn´s disease, intraabdominal abscess, penetrating ileitis, microbial spectrum, antibiotic resistance

## Abstract

Intraabdominal abscess formation occurs in up to 30% of patients suffering from Crohn´s disease (CD). While international guidelines recommend a step-up approach with a combination of empiric antibiotic therapy and percutaneous drainage to delay or even avoid surgery, evidence about microbial spectrum in penetrating ileitis is sparse. We retrospectively assessed outcomes of 46 patients with terminal penetrating Ileitis where microbial diagnostics have been performed and compared microbial spectrum and antibiotic resistance profile of CD patients with patients suffering from diverticulitis with intraabdominal abscess formation. In both groups, the most frequently isolated pathogen was the gram-negative bacterium *E. coli* belonging to the family of Enterobacterales. However, overall Enterobacterales were significantly more often verifiable in the control group than in CD patients. Furthermore, microbial analysis showed significant differences regarding isolation of anaerobic pathogens with decreased frequency in patients with CD. Subgroup analysis of CD patients to evaluate a potential influence of immunosuppressive therapy on microbial spectrum only revealed that *Enterobacterales* was less frequently detected in patients treated with steroids. Immunosuppressive therapy did not show any impact on all other groups of pathogens and did not change antibiotic resistance profile of CD patients. In conclusion, we were able to demonstrate that the microbial spectrum of CD patients does differ only for some pathogen species without increased rate of antibiotic resistance. However, the empiric antibiotic therapy for CD-associated intra-abdominal abscess remains challenging since different points such as local epidemiological and microbiological data, individual patient risk factors, severity of infection, and therapy algorithm including non-surgical and surgical therapy options should be considered before therapeutical decisions are made.

## 1. Introduction

Crohn´s disease (CD) is a chronic and relapsing inflammatory bowel disease characterized by transmural inflammation which can result in fistulae and intraabdominal abscess formations [1,2]. Therapeutic strategies for intraabdominal abscesses that occurs in up to 30% of patients suffering from Crohn´s disease include antibiotic treatment, interventional drainage, and surgical approaches [3,4,5,6,7,8]. International guidelines recommend a step-up approach with a combination of (empiric) antibiotic therapy and percutaneous CT- or ultrasound-guided drainage, if technically possible, to delay or even avoid surgery for intestinal resection [6,9,10,11]. However, this therapeutical concept is rather good clinical practice than evidence-based medicine since high-quality studies such as prospective randomized studies are still lacking. Furthermore, the benefit of percutaneous drainage is still in discussion as recently published studies have presented conflicting results [12,13,14]. Therefore, antibiotics still play a fundamental role in intra-abdominal abscess formations in CD to prevent further septic complications. Given that antibiotic treatment should not be delayed after identification of intra-abdominal abscess and interventional drainage is not always technically feasible, empiric antibiotic therapy should consider the potential spectrum of pathogens, individual patient risk factors, severity of infection, and local epidemiology including potential antibiotic resistances. There are studies showing *E. coli*, belonging to the group of Enterobacterales, *Streptococcus* spp., *Enterococcus* spp., anaerobes, and *Candida* spp. as the most causative pathogens for intraabdominal abscesses in CD patients [3,15,16]. Additionally, these data demonstrate an increasing appearance of multi-resistant bacterial strains.

As the available evidence remains weak, further clinical studies are needed to strengthen therapeutic recommendations.

Thus, the objective of this single-center retrospective study was to investigate and evaluate the microbial spectrum, potential antibiotic resistance, and influence of immunosuppressive therapy in CD patients with intra-abdominal abscess formations compared to patients suffering from diverticulitis associated abscess findings.

## 2. Materials and Methods

### 2.1. Study Population

In this single-center retrospective study, all patients with ileocecal resection due to Crohn’s disease with terminal penetrating ileitis (Montreal classification L1B3) [17] and consecutive intra-abdominal abscess formation treated from 1 January 2012 to 31 December 2019 at the Department of Surgery at the University Hospital of Wuerzburg were evaluated. The control group did contain patients with sigmoid diverticulitis and intra-abdominal abscess who underwent surgery between 1 January 2014 and 31 December 2019. In total, 46 CD patients and 50 patients with sigmoid diverticulitis were identified. Patients without microbial diagnostics were excluded (Figure 1).

Terminal ileitis was defined as inflammation limited to the terminal ileum with the histopathological diagnosis of Crohn’s disease. Abscess formation was diagnosed either pre-operatively by computer tomography (CT)/magnetic resonance imaging (MRI) or intra-operatively by a surgeon. Patient´s characteristics including medical history, pre-existing conditions (comorbidities), and laboratory parameters were collected for each patient from patient records.

### 2.2. Microbial Diagnostic

For microbiological diagnostic reasons, diagnostic swabs from each abscess formation were performed during surgery or secretion was gained by percutaneous drainage. Microbial diagnostic including microbial cultures was processed and analyzed in the Institute of Hygiene and Microbiology at the University of Wuerzburg, in accordance with national and international standards [18]. Pathogens were divided into the following groups: gram-positive Streptococci including viridans and beta-hemolytic Streptococci, Staphylococci, Enterobacterales, Enterococci, Fungi, and clinically non relevant pathogens (Appendix A).

To be able to interpret the microbial results in the context, we used local epidemiological data including antibiotic resistances and antibiotic usage. These data are routinely collected by the in-house antibiotic stewardship team of the University Hospital Wuerzburg.

### 2.3. Subgroup Analysis

In a subgroup analysis of all CD patients, the microbial spectrum was analyzed with respect to immunosuppressive therapy (no therapy, steroid-based immunosuppressive therapy, and antibody-based immunosuppressive therapy).

### 2.4. Statistical Analysis

Descriptive data were presented as median with range or total numbers with percentage. Differences in patient characteristics were assessed by Chi-Square test, Fisher’s exact test, or ANOVA test in accordance with the data scale and distribution. A *p*-value of <0.05 was considered statistically significant. Statistical analysis was done by using SPSS statistics (Version 25, IBM, Armonk, NY, USA).

### 2.5. Ethical Considerations

Ethical approval for this study was obtained from the Ethics Committee of the University of Wuerzburg, Germany.

## 3. Results

### 3.1. Patient Characteristics

In this retrospective single-center study, 104 patients age 16 years or older with penetrating ileitis and intra-abdominal abscess formation who underwent surgical resection were initially identified. After exclusion of 58 patients due to missing microbial diagnostic, 46 CD patients and 50 patients operated due to diverticulitis (control group) were finally included (Figure 1). While patients suffering from Crohn´s disease were significantly younger (40.4 vs. 64.9 years; *p* < 0.001) and characterized by lower body mass index (23.6 vs. 28.0; *p* < 0.001) compared to the control group, both groups did not show any significant differences regarding gender, comorbidities (cardiovascular, pulmonary, diabetes), ASA (American Society of Anesthesiologists) classification, and history of smoking. Pre-operative laboratory testing revealed a significant higher C-reactive protein in the control group (10.3 vs. 6.0 mg/dL; *p* < 0.033). Values for hemoglobin, albumin, and leukocytes were comparable without any significant differences (Table 1).

### 3.2. Microbial Outcome

Microbial analysis revealed sterile cultures in 12 patients (12.6%) (Table 2). Here, patients with CD (*n* = 10) showed significantly more often a negative result in comparison to the control group (21.7 vs. 4.0%; *p* = 0.012). In the remaining 36 patients suffering from CD, a total number of 77 pathogens could be isolated resulting in a median of 2 pathogens per culture (range: 1–6). While microbial culture in 2 patients of the control group was negative, 145 different pathogens were detected in the remaining 48 patients with a median of 3 pathogens per culture (range: 1–9). In both groups, the most frequently isolated pathogen was gram-negative bacterium *E. coli* belonging to the family of *Enterobacterales.* However, overall *Enterobacterales* were significantly more often verifiable in the control group than in CD patients (74.0 vs. 43.5%; *p* = 0.003). The most frequently isolated gram-positive species were *Streptococcus* spp. (32.0 vs. 34.8%) and *Enterococcus* spp. (30.0 vs. 15.2%; *p* = 0.096) without any significant differences between both groups. Fungal pathogens could be detected in 10 CD patients (21.7%) and 9 patients of the control group (18.0%), respectively. Microbial analysis showed significant differences regarding isolation of anaerobic pathogens. Here, anaerobes were more frequently isolated in control patients compared to patients with CD (62.0 vs. 19.6%). In summary, abscess formations in patients suffering from penetrating ileitis due to CD were characterized by significantly lower detection of *Enterobacterales* and anaerobes compared to the control group (Table 2).

### 3.3. Microbial Spectrum and Influence of Immunosuppressive Therapy

To investigate the potential impact of immunosuppressive medication on the microbial spectrum isolated from intra-abdominal abscesses, all CD patients were divided into three groups: no immunosuppressive therapy, steroid-based immunosuppressive therapy, and antibody-based immunosuppressive therapy (Table 3). Here, a trend could only be detected for *Enterobacterales* and patients treated with steroids resulting in decreased occurrence in intra-abdominal abscesses (25%; *p* = 0.108). For all other groups of pathogens, immunosuppressive therapy did not show any impact on microbial spectrum.

### 3.4. Empiric Antibiotic Therapy and Resistance Profile

Even if microbial diagnostic revealed that the occurrence of Enterobacterales was less frequent in CD patients compared to the control group, the gram-negative bacterium *E. coli* was the most isolated pathogen in both groups. Therefore, we evaluated the initial empiric antibiotic therapy of all patients and its effectiveness against *E. coli*. *E. coli* was isolated by bacterial culture in 15 CD patients of whom 14 patients received an empirical antibiotic therapy. Antibiotic regime was individually chosen by the attending physician. While five patients were treated with ciprofloxacin, six patients received cephalosporines (1 × 1st generation (cefazolin), 2 × 2nd generation (cefuroxime), and 3 × 3rd generation (ceftriaxone)). A broad-spectrum antibiotic was only administrated in two patients (1× meropenem, 1× piperacillin/tazobactam). Except in one patient who was treated by monotherapy with metronidazole, the different empirical antibiotic therapies were sufficient for the treatment of all isolated *E. coli* species (Table 4).

### 3.5. Local Epidemiological Data and Antibiotic Resistances

As shown in Appendix A the resistance rates of *E. coli* against the third generation cephalosporine cefotaxime and ciprofloxacin have continuously decreased after 2014. In parallel, the oral and intravenous administration of these two antibiotics could be reduced (Appendix A).

## 4. Discussion

Fistulizing Crohn´s disease is clinically demanding and requires a multidisciplinary approach. International guidelines recommend intravenous antibiotic therapy and therapeutic (percutaneous) drainage if abscess formation exceeds 3 cm in diameter [6,9,10,11]. So far, empiric antibiotic therapy included a combination of metronidazole with fluoroquinolones or third generation cephalosporins [6,18], but these recommendations are mainly based on case series and retrospective monocentric data, respectively [15,19,20,21,22,23,24,25,26,27,28,29,30,31]. In line with that, a recently published multicentric prospective European study evaluating the microbial spectrum of intra-abdominal abscess formation in patients suffering from Crohn´s disease demonstrated a high rate of inadequate antimicrobial empirical first-line therapy [16]. Thus, further evidence is necessary to extend current knowledge and to gain more robust microbiological data on pathogens in CD-associated intra-abdominal abscesses. Importantly, while antimicrobial stewardship programs are currently evolving, the prescription of fluoroquinolones and third generation cephalosporins such as ceftriaxone or cefotaxime must be questioned and, if feasible, substituted for less unfavorable agents given the increasing prevalence of resistance among Enterobacterales (e.g., ESBL). This is supported by national and international safety warnings on both antibiotic classes based on potent adverse events of fluoroquinolones [32,33,34].

In this monocentric retrospective study, microbial spectrum of CD patients suffering from intra-abdominal abscess were evaluated compared to patients treated due to diverticulitis-associated abscess formations. Interestingly, microbial analysis revealed that Enterobacterales and anaerobes were less frequently isolated in CD patients, although it has been widely reported that *E. coli* may play an important role in pathogenesis of CD [19,35]. However, even if the detection of Enterobacterales species was significantly decreased compared to the control group, *E. coli* was the most frequently isolated pathogen in both groups. This observation is supported by data from a German multicentric prospective register study showing E. coli and *Streptococcus* spp. as the most frequently detectable pathogens in microbial cultures of CD-associated intra-abdominal formations [16]. Interestingly, microbial analysis in CD patients was more often characterized by sterile cultures, which might be a result of longer empiric antibiotic therapy. Here, further clinical studies are necessary to confirm this result and to investigate potential pathomechanisms.

As mentioned above, international guidelines recommend an antibiotic therapy combination with metronidazole to cover anaerobic pathogens. However, in our study this group of pathogens was only verifiable in less than 20% of CD patients, reflecting the results of other studies. However, a positive rate of 20% is not debatable and therefore metronidazole should be generally recommended for first-line antibiotic therapy.

As shown in our study as well as in the literature, a broad spectrum of different antimicrobial therapies is used in clinical routine due to missing robust epidemiological and microbiological data on pathogen patterns in CD-associated intra-abdominal abscess formations [16]. Given that, it must be carefully considered that there is a growing body of evidence demonstrating an increasing risk of antibiotic resistance in patients suffering from intra-abdominal abscess [15,16,26,36,37]. Non-sufficient empirical first-line therapy is associated with extended hospital stay and higher mortality resulting in increased healthcare-related costs [38,39,40,41,42,43]. Furthermore, there was no evidence for an increased risk of multidrug-resistant pathogens despite an immunosuppressive therapy in our cohort. Empiric antibiotic therapy for the most frequently isolated pathogen, *E. coli*, was sufficient except for one patient who was treated with a monotherapy of metronidazole. The occurrence and development of multidrug-resistant pathogens depend on individual patient risk factors including antibiotic pre-treatments, severity of infection, and local epidemiology. As shown in Appendix A, local epidemiological data from the tertiary hospital center of the University Hospital of Wuerzburg indicate ciprofloxacin and cefotaxime (ceftriaxone) resistance rates between 10% and 26% of *E. coli*, suggesting that these antibiotic agents may work as first-line therapy, provided antibiotic resistance of Enterobacterales remains below 20%. Our in-house antibiotic stewardship team managed since its implementation 2015 to reduce the antibiotic consumption on third generation cephalosporins and fluoroquinolones in collaboration with our surgical department successfully, resulting in fewer resistance rates (Appendix A). Furthermore, this analysis also includes samples from other high-risk populations such as patients suffering from hematological and oncological diseases who are often characterized by colonization with multidrug-resistance pathogens due to advanced therapies and long hospital stays with potential nosocomial infections and urgent need for antibiotic broad-spectrum therapies. Therefore, continuous evaluation and surveillance of local epidemiological microbial data are mandatory to enable adaptation of empiric antibiotic therapy. It has been recently shown that the implementation of antimicrobial stewardship programs is a highly effective tool to provide these data mentioned above and to supervise antibiotic treatment in intra-abdominal infections [44,45,46,47].

Common recommendations for first-line therapy in intra-abdominal abscess do not enclose antifungal medication, but recently published studies and our results showed a notable isolation rate of fungi species [16,26,48]. Intra-abdominal infection with Candida has been reported for patients hospitalized on intensive care units and for infections originated in the upper gastrointestinal tract due to physiological colonization [49,50]. Candida-associated infection are usually associated with poor prognosis accompanied by a high mortality which further increases if antifungal therapy is delayed [50,51,52]. Given that most CD patients are immunocompromised due to medical therapy, it is surprising that the percentage of antifungal empiric therapy was zero percent in our study and in the multicenter study of Reuken et al. which also evaluated the microbial spectrum of intraabdominal abscesses in perforating Crohn’s disease using data from a prospective German registry [16]. However, the lack of antifungal therapy seems to play no pivotal role regarding clinical outcome; thus, further studies are necessary to address the role of antifungal therapy as first-line medication in intraabdominal abscess formation.

Since many patients suffering from Crohn´s disease receive an immunosuppressive therapy, there is always the question if an empiric antibiotic therapy should be adopted. Subgroup-analysis revealed no significant differences of pathogens with respect to immunosuppressive therapy except for patients treated with steroids who were characterized by lower detection of Enterobacterales. This is in contrast to the study published by Reuken et al., where gram-negative infection was significantly more frequent in patient with steroid-based therapy [16].

Our study has some important limitations that merit acknowledgment. The retrospective monocentric character of our study attenuates the significance of our results and the transferability to other hospital centers. Furthermore, the number of patients included limits conclusions and lessons learned from this study. Additionally, retrospective analysis did not enable to show antimicrobial susceptibility of all isolated pathogens. Here, further prospective studies are necessary to study this clinically relevant issue in more detail. However, this is the first study comparing the microbial spectrum of intra-abdominal abscess formations from CD and non-CD patients.

## 5. Conclusions

Empiric antibiotic therapy for CD-associated intra-abdominal abscess remains challenging since different aspects such as local epidemiological and microbiological data, individual patient risk factors, severity of infection, and therapy algorithm including non-surgical and surgical therapy options should be considered before therapeutical decisions are made. We were able to demonstrate that the microbial spectrum of CD patients does differ only for some pathogen species without increased rate of antibiotic resistance. Thus, empiric antimicrobial treatment without broad-spectrum antibiotic and antifungal drugs is a reasonable therapy in CD patients suffering from penetrating ileitis. However, it should be considered that epidemiological data depend on local circumstances. Therefore, antimicrobial stewardship could help to control local development of potential microbial resistances and to supervise empiric antibiotic therapies aiming towards sufficient and best available care for CD patients.

## Figures and Tables

**Figure 1 jcm-11-04343-f001:**
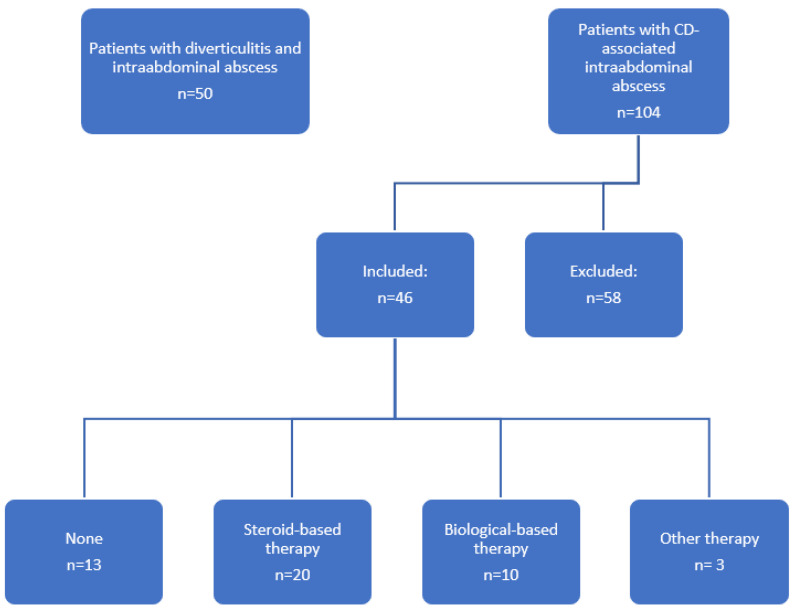
Study design.

**Table 1 jcm-11-04343-t001:** Patient demographics.

Variable	Control (*n* = 50)	Crohn’s Disease (*n* = 46)	*p*
Age (year)			
Median	64.9	40.4	<0.001
(range)	(19.7–88.6)	(16.6–66.9)	
Sex (*n*,%)			
Female	25 (50.0)	15 (32.6)	
Male	25 (50.0)	31 (67.4)	0.100
BMI (kg/m^2^)			
Median	28.0	23.6	<0.001
(range)	(17.5–47.1)	(13.6–37.0)	
CVRF (*n*,%)	33 (66.0)	22 (47.8)	0.099
COPD (*n*,%)	8 (16.0)	3 (6.5)	0.203
Diabetes (*n*,%)	1 (2.0)	3 (6.5)	0.347
Smoking			
active (*n*,%)	7 (14.0)	10 (21.7)	
former (*n*,%)	2 (4.0)	1 (2.2)	0.557
ASA classification (*n*,%)			
1	1 (2.0)	2 (4.3)	
2	25 (50.0)	28 (60.9)	
3	22 (44.0)	14 (30.4)	
4	2 (4.0)	2 (4.3)	0.548
Hb (g/dL)			
median (range)	12.9 (5.3–17.6)	11.9 (6.6–19.8)	0.530
Albumin (g/L)			
median (range)	4.0 (2.2–5.0)	3.7 (2.4–5.1)	0.722
Leukocytes (×1000/µL)			
median (range)	13.5 (4.5–30.2)	12.8 (4.4–43.1)	0.375
CRP (mg/dL)			
median (range)	10.3 (0.1–38.2)	6.0 (0.2–34.8)	0.033

ASA, American Society of Anesthesiologists; BMI, body mass index; COPD, chronic obstructive pulmonary disease; CRP, C reactive protein; CVRF, cardiovascular risk factor; Hb, hemoglobin.

**Table 2 jcm-11-04343-t002:** Microbial spectrum of intra-abdominal abscesses.

Pathogen	Control (*n* = 50)	Crohn’s Disease (*n* = 46)	*p*
Negative (*n*,%)	2 (4.0)	10 (21.7)	0.012
Skin germs (*n*,%)	0 (0)	2 (4.3)	0.227
Viridans group streptococci (*n*,%)	14 (28.0)	15 (32.6)	0.661
Streptococcus groups A, B, C and G (*n*,%)	2 (4.0)	1 (2.2)	1.000
*Staphylococcus* spp. (*n*,%)	4 (4.0)	2 (4.3)	1.000
Anaerobic bacteria (*n*,%)	31 (62.0)	9 (19.6)	0.001
*Enterobacterales* (*n*,%)	37 (74.0)	20 (43.5)	0.003
*Enterococcus* spp. (*n*,%)	15 (30.0)	7 (15.2)	0.096
Non relevant (*n*,%)	2 (4.0)	4 (8.7)	0.422
Fungi (*n*,%)	9 (18.0)	10 (21.7)	0.798

**Table 3 jcm-11-04343-t003:** Microbial spectrum and influence of immunosuppressive therapy.

Pathogen	No Immunosuppressive Therapy (*n* = 13)	Steroid-Based Immunosuppressive Therapy (*n* = 20)	Antibody-Based Immunosuppressive Therapy (*n* = 10)	*p*
Negative (*n*,%)	1 (7.7)	7 (35.0)	2 (20.0)	0.186
Skin germs (*n*,%)	2 (15.4)	0 (0)	0 (0)	0.089
Viridans group streptococci (*n*,%)	5 (38.5)	7 (35.0)	3 (30.0)	0.915
Streptococcus groups A, B, C and G (*n*,%)	1 (7.7)	0 (0)	0 (0)	0.307
*Staphylococcus* spp. (*n*,%)	1 (7.7)	1 (5.0)	0 (0)	0.682
Anaerobic bacteria (*n*,%)	2 (15.4)	5 (25.0)	1 (10.0)	0.572
*Enterobacterales (*n*,%)*	7 (53.8)	5 (25.0)	6 (60.0)	0.108
*Enterococcus* spp. (*n*,%)	2 (15.4)	1 (5.0)	2 (20.0)	0.424
Non relevant (*n*,%)	0 (0)	2 (10.0)	0 (0.)	0.299
Fungi (*n*,%)	2 (15.4)	6 (30.0)	1 (10.0)	0.376

**Table 4 jcm-11-04343-t004:** Empiric antibiotic therapy and resistance profile for *E. coli*.

Antibiotic Regime	Number (*n*)	Sensible	Intermediary	Resistant	No Therapy
Overall	14	11	2	1	1
CIP/MET	5	5			
CEFT/MET	3	3			
CEFU/MET	2	1	1		
CEFA/MET	1		1		
MERO	1	1			
MET	1			1	
PIP/TAZ	1	1			

CIP, ciprofloxacin; MET, metronidazole, CEFT, ceftriaxone; CEFU, cefuroxime; CEFA, cefazoline; MERO, meropenem; PIP/TAZ, piperacillin/tazobactam.

## Data Availability

Institutional database. Therefore, restrictions to availability apply due to data protection regulations. Anonymized data are, however, available from the corresponding author on reasonable request.

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
