# Peer review of "Microbial Spectrum and Antibiotic Resistance in Patients Suffering from Penetrating Crohn’s Disease"

_jcm, 2022, doi:10.3390/jcm11154343_

Round 1

Reviewer 1 Report

This is an interesting manuscript that provides data on microbiology in patients with Crohn’s disease and patients without Crohn’s disease that also had intraabdominal abscesses. I have some comments, the most important of them being the one regarding data on antimicrobial resistance that should be provided in this specific patient population. Please find some comments below:

1.       Line 111: Please define all abbreviations when first used

2.       Table 1: All abbreviations should be defined as a footnote

3.       Line 152: Why were there sterile cultures more often in the CD group? Do you have any data regarding the days or the type of antimicrobial treatment before obtaining the cultures?

4.       Table 2: Just to make it clear: Do the percentages here refer to all patients, including the ones with a negative culture, or do they refer to those with a positive culture? I am not quite sure which analysis would be more appropriate. I guess the second

5.       Line 174: Please state that it is a trend, as the difference is not statistically significant

6.       As also stated in a previous comment, I would be interested to see if there are any differences among different treatment groups and previous treatment

7.       Do you have any data on antimicrobial susceptibility? That would be interesting to see. It would also allow seeing if the antimicrobial treatment that was administered to the CD patients was adequate or not

8.       Were there any differences in the two groups in the size of the abscesses?

9.       Line 243: I feel that whether an antimicrobial with activity against anaerobes should be administered is not quite debatable. 20% is quite high to ignore

10.   Line 254: Here the authors say that antimicrobial treatment was adequate, however, I did not see that in the results section

11.   Line 260: The authors here provide some data that are shown in the supplementary data, but this should be done in the results, not in the discussion section

12.   I understand that the authors provide some data on antimicrobial susceptibility from their institution in general; however, antimicrobial resistance in CD patients may differ, as these patients may be more heavily treated with antimicrobials and are immunocompromized, thus, they could be more frequently colonized by resistant microorganisms

13.   Line 289: Again, in order to say that, you have to provide data on antimicrobial resistance in the specific study population. Providing the species names is not enough

Author Response

Dear Dr. Liu,

Dear Reviewers,

Thank you for giving us the opportunity to reply to your comments on our manuscript entitled “Microbial spectrum and antibiotic resistance in patients suffering from penetrating Crohn´s disease”.

We appreciate the time and effort that you and the reviewers have dedicated to providing us your valuable feedback on our manuscript. We are grateful to the reviewers for their insightful comments on this paper. Please see below a point-by-point response to the reviewers’ comments and concerns. All changes in the manuscript are highlighted using the “Track Changes” function.

Comments from Reviewer 1:

1. Line 111: Please define all abbreviations when first used.

Answer: We thank the reviewer for this advice, and we defined all abbreviations when first used.

  1. Table 1: All abbreviations should be defined as a footnote.

Answer: We thank the reviewer for this advice and added footnotes.

  1. Line 152: Why were there sterile cultures more often in the CD group? Do you have any data regarding the days or the type of antimicrobial treatment before obtaining the cultures?

Answer: We appreciate this comment since this is a very important point requiring further studies. Based on the German health care system, many CD patients are treated independently in gastroenterological outpatient clinics and are presented to surgical departments when non-surgical treatment is not longer successful. Thus, we do not have sufficient data from all patients regarding antibiotic pre-treatment. However, we believe that CD patients have received a longer empiric antibiotic therapy resulting in higher rate of sterile cultures. However, we are planning to address this clinically relevant issue in a new research project. We added the following sentences in the Discussion: “Interestingly, microbial analysis in CD patients was more often characterized by sterile cultures, which might be a result of longer empiric antibiotic therapy. Here, further clinical studies are necessary to confirm this result and to investigate potential pathomechanism.”

  1. Table 2: Just to make it clear: Do the percentages here refer to all patients, including the ones with a negative culture, or do they refer to those with a positive culture? I am not quite sure which analysis would be more appropriate. I guess the second

Answer: We thank the reviewer for this comment. The percentage refer to all patients including the ones with sterile culture, since the aim of our study was to show the microbial spectrum and the percentage of different microbes of all patients suffering from penetrating CD or diverticulitis with abscess formation. Therefore, we decided to show the percentage referring to all patients.

  1. Line 174: Please state that it is a trend, as the difference is not statistically significant.

Answer: We thank the reviewer for this advice, and we changed the sentence.

  1. As also stated in a previous comment, I would be interested to see if there are any differences among different treatment groups and previous treatment.

Answer: We thank the reviewer again for this clinical important question and we would like to refer to our answer 3. Since previous history of antimicrobial therapy is not sufficiently recorded for all patients, especially the duration of treatment, we are not able to answer this question with our patient cohort. As we mentioned already above, we currently address this issue in a new research project.

  1. Do you have any data on antimicrobial susceptibility? That would be interesting to see. It would also allow seeing if the antimicrobial treatment that was administered to the CD patients was adequate or not.

Answer:  We would like to thank the reviewer for this comment. This is indeed a point that ought not to be neglected. Unfortunately, we do not have any data available on the antimicrobial susceptibility of the patients recruited into this study but given the resistance profile in the department of general surgery the administered antibiotics were most likely to be appropriate. Depending on the expected source and given resistance profile on the potentially responsible pathogens the antibiotic regime is chosen. But as we were planning on a sequel, we would like to give this aspect more evidence.

  1. Were there any differences in the two groups in the size of the abscesses?

Answer: We do not have data about the size of the abscess formation, since detection of abscess was also performed intraoperatively, and thus, no sufficient imaging for measurement was available.

  1. Line 243: I feel that whether an antimicrobial with activity against anaerobes should be administered is not quite debatable. 20% is quite high to ignore

Answer: Yes, we agree. Anaerobic bacteria should be covered as it is so in our department, we modified accordingly.

  1. Line 254: Here the authors say that antimicrobial treatment was adequate, however, I did not see that in the results section.

Answer: The results for the antimicrobial treatment are presented in section 3.4: empirical antibiotic therapy and resistance profile. Here we focused on E.coli since this bacterium was the most frequently isolated pathogen in our patients. To better visualize these results we added table 4.

  1. Line 260: The authors here provide some data that are shown in the supplementary data, but this should be done in the results, not in the discussion section.

Answer: We thank the reviewer for this advice. We have changed the manuscript and now present these results in the results section.

  1. I understand that the authors provide some data on antimicrobial susceptibility from their institution in general; however, antimicrobial resistance in CD patients may differ, as these patients may be more heavily treated with antimicrobials and are immunocompromized, thus, they could be more frequently colonized by resistant microorganisms.

Answer: We appreciate this important question, but essentially the most cultured pathogen has been E. coli, and the resistance profile did not show an increased resistance as shown in figure 4. Our aim was to investigate the microbial spectrum of CD patients in general, but we also wanted to show that there is no increased risk of multi-resistant pathogens, even if these population might be characterized by immunosuppression and antimicrobial therapy.  

Furthermore, due to the polymicrobic nature not each of the organisms will generally identified or are provided with an antibiogram, on special request with given risk factors (e.g.immunosuppression) certain pathogens will be named and differentiated by resistance.

  1. Line 289: Again, in order to say that, you have to provide data on antimicrobial resistance in the specific study population. Providing the species names is not enough.

Answer: As we already mentioned above (point 10 and 12), we focused on E.coli, since this pathogen was the most frequently isolated in our patients, and secondly E. coli is known to be able to develop antibiotic resistance and is responsible for nosocomial infections. Given that, we could demonstrate that CD patients with E.coli are not characterized with a higher rate of antibiotic resistance as shown in figure 4.

Additionally, we could isolate MRSA only in one CD patients, thus, we did not want to draw a conclusion from this observation. In summary, we think, that empiric antibiotic therapy in CD patients can be adopted to patients suffering from diverticulitis, if local epidemiological circumstances are properly considered.

If you require any further information, please do not hesitate to contact me. Thank you in advance.

Sincerely

Sven Flemming, MD

Head of Surgery for Inflammatory Bowel Disease

Department of General, Visceral, Transplantation, Vascular and Pediatric Surgery; Center of Operative Medicine (ZOM), University Hospital of Wuerzburg, Wuerzburg, Germany

Reviewer 2 Report

1)Lines 38-39 are better developed IBD globally, not only among young people but also among pediatrics, and explain more concern about the treatment. please add this reference The emerging epidemic of inflammatory bowel disease in Asia and Iran by 2035: A modeling study DOI: 10.1186/s12876-021-01745-1

2) Do line 83-84 need references and explain more about the bacterial detection method?

3) why did not do an antibiogram on the result of the samples?

4) what are your criteria for choosing different antibiotics for treatment?

5) are you follow up with the patients? what is the result of treatment explained more clearly?

Author Response

Dear Dr. Liu,

Dear Reviewers,

Thank you for giving us the opportunity to reply to your comments on our manuscript entitled “Microbial spectrum and antibiotic resistance in patients suffering from penetrating Crohn´s disease”.

We appreciate the time and effort that you and the reviewers have dedicated to providing us your valuable feedback on our manuscript. We are grateful to the reviewers for their insightful comments on this paper. Please see below a point-by-point response to the reviewers’ comments and concerns. All changes in the manuscript are highlighted using the “Track Changes” function.

Comments from Reviewer 2:

  1. Lines 38-39 are better developed IBD globally, not only among young people but also among pediatrics, and explain more concern about the treatment. please add this reference: The emerging epidemic of inflammatory bowel disease in Asia and Iran by 2035: A modeling study DOI: 10.1186/s12876-021-01745-1 

Answer: We thank the reviewer for this comment. However, the aim of our study was not to describe epidemiological data about the global burden of Crohn´s disease and its immunosuppressive therapy. Our focus was the microbial spectrum of intraabdominal abscess formation resulting from penetrating ileitis in CD patients.

  1. Do line 83-84 need references and explain more about the bacterial detection method?

Answer: Many thank for this meaningful comment. We added the reference and hope it has become clearer that the detection methods are in accordance to national and international standards

  1. Why did not do an antibiogram on the result of the samples?

Answer: Thank you for raising up this important point. As mentioned above for reviewer 1 we have focused in E.coli as the most frequently isolated pathogen and evaluated here the resistance profiles. However, we are planning further prospective studies to address this clinically relevant issue in more detail.

  1. What are your criteria for choosing different antibiotics for treatment?

Answer: The choice of antimicrobial therapy was in the hand of the physician in charge. Thus, the therapy based on individual preferences and knowledge. However, to address this problem our in-house antimicrobial stewardship team provided a local guideline on surgical and non-surgical infections, launched 2020.

  1. Are you follow up with the patients? what is the result of treatment explained more clearly?

Answer: We only have a regular follow-up of 30 days after operation. Here, we did not see any events related to the initial antimicrobial therapy.

If you require any further information, please do not hesitate to contact me. Thank you in advance.

Sincerely

Sven Flemming, MD

Head of Surgery for Inflammatory Bowel Disease

Department of General, Visceral, Transplantation, Vascular and Pediatric Surgery; Center of Operative Medicine (ZOM), University Hospital of Wuerzburg, Wuerzburg, Germany

Round 2

Reviewer 1 Report

The manuscript has been improved during the revision process. However, since not presenting data on antimicrobial susceptibility is a limitation, it should be mentioned in the limitations subsection of the discussion section.
